# The Impact of Drought on Soil Moisture Trends across Brazilian Biomes

[1]Flavio Lopes Ribeiro, [2]Mario Guevara, [2]Alma Vázquez-Lule, [3]Ana Paula Cunha, [3]Marcelo Zeri, [2]Rodrigo Vargas

[1]University of Delaware, School of Public Policy and Administration, Disaster Research Center, Newark, DE, USA
[2]University of Delaware, Department of Plant and Soil Sciences, Newark, DE, USA
[3]National Center for Monitoring and Early Warning of Natural Disasters (CEMADEN), São José dos Campos, SP, Brazil

*Correspondence to*: Flavio Lopes Ribeiro

**Abstract**: Over the past decade, Brazil has experienced severe droughts across its territory, with important implications for soil moisture dynamics. Soil moisture variability has a direct impact on agriculture, water security, and ecosystem services. Nevertheless, there is currently little information on how soil moisture across different biomes respond to drought. In this study, we used satellite soil moisture data from the European Space Agency, from 2009 to 2015, to analyze differences in soil moisture responses to drought for each biome of Brazil: The Amazon, Atlantic Forest, Caatinga, Cerrado, Pampas and Pantanal. We found an overall soil moisture decline of -0.5%/year (p<0.01) at the national level. At the biome-level, Caatinga presented the most severe soil moisture decline (-4.4% per year); whereas Atlantic Forest and Cerrado biomes showed no significant trend. The Amazon biome showed no trend but a sharp reduction of soil moisture from 2013 to 2015. In contrast, Pampas and Pantanal presented a positive trend (1.6 and 4.3 %/year, respectively). This information provides insights for drought risk reduction and soil conservation activities to minimize the impact of drought in the most vulnerable biomes. Furthermore, improving our understanding of soil moisture trends during periods of drought is crucial to enhance the national drought early warning system and develop customized strategies for adaptation to climate change in each biome.

## 1. Introduction

Drought is a natural and human-induced hazard common to all climate zones in the world (Sheffield and Wood, 2008), generally referred to as a sustained occurrence of below average water availability due to precipitation deficit and soil moisture decline (Magalhães, 2016). Precipitation deficit is the most studied driver of drought (Mishra and Singh 2010; Smith 2013, Villarreal et al., 2016) and has been furthering several drought indicators and models. However, precipitation-based indicators are limited in the assessment of social and environmental responses to the lack of rain and therefore not suitable for evaluating the impacts of drought when used alone. On the other hand, drought indicators based on soil moisture are not only key to understanding the physical mechanisms of drought, but also useful for assessing how soil moisture decline can alter vegetation water availability and, consequently, agricultural production and ecosystem services (Smith 2013; NWS 2008).

When soil moisture declines below critical water stress thresholds it reduces biomass production, soil respiration
and the overall soil carbon balance (Bot and Benites 2005; Vargas et al., 2018). Low carbon in soils (due to lower
biological activity) reduces its structural integrity and increases the risk of soil erosion, contributing to river silting,
ineffective runoff control, and loss of soil nutrients (Al-Kaisi and Rattan 2017). Soil moisture is also crucial for
addressing the negative impacts of climate change in water and land resources (Bossio 2017). Indeed, temporal
variability of soil moisture in a given biome is an important variable for the characterization of the local climate
(Legates et al. 2011) and a key indicator of changes in the biome's water cycle (Sheffield and Wood 2008; Rossato
et al. 2017).
In this study, we use satellite data from the European Space Agency (ESA) to analyze the impact of drought on
soil moisture across all Brazilian biomes: The Amazon, Atlantic Forest, Caatinga, Cerrado, Pampas and Pantanal.
Considering that each biome has distinct climate, soil and vegetation characteristics, we hypothesize that they
would respond differently to drought conditions (e.g., positive, negative or non-significant) and show up relevant
information for drought management at national and regional levels.
In Brazil, most of the work on drought management has been focused in the semiarid region, well-known for its
recurrent problems with droughts and water scarcity (Fig. 1) and where predominates the Caatinga biome.
However, droughts have been reported all over Brazil, affecting all other biomes as well. In the period selected
for this study (i.e., 2009 to 2015), there was a high number of municipalities declaring emergency and even public
calamity due to drought across the country (Cunha et al. 2019), but the impacts on soil moisture at national scale
and how each biome responds to drought are still unknown.

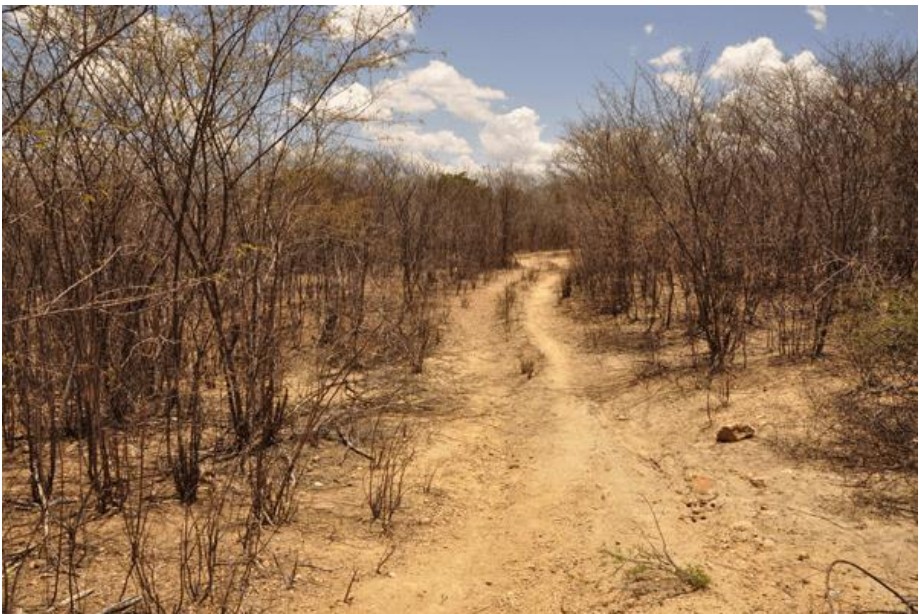

**Figure 1: The Caatinga biome (Pontes, 2012).**
Due to climate change, extreme events such as drought are expected to become more intense and recurrent in
some regions of Brazil. Therefore, integrating satellite soil moisture data into early warning systems could
contribute to more efficient drought risk management and promote data-driven climate change adaptation.
Nevertheless, studies on soil moisture variation have been conducted at a stand-scale due to challenges for
measurements across spatial and temporal scales (Legates et al. 2011; Novick et al 2016). As a consequence, the
lack of soil moisture information could lead to inaccurate assessment of drought conditions, underestimation of
drought impacts, and incomplete resilience and adaptation plans. As droughts become more frequent and intense,
it is important to enable monitoring of soil moisture trends and communicate the results at different levels (e.g.,
municipal, state, national, regional) and across different perspectives (e.g., environmental, social, and economic).
At present, the most reliable source of soil moisture information at large-scales (i.e., global-to-continental scales)
is satellite remote sensing (i.e., https://smap.jpl.nasa.gov/, http://www.esa-soilmoisture-cci.org/), which provides
soil moisture estimates for the first 0-5 cm of soil depth (Liu et al. 2011). Even though the first layer of soil is
expected to be very dynamic because of its interaction with the atmosphere and deeper layers still represent an
important water storage, especially in the Amazon and Cerrado biomes, soil moisture at the first 5cm is still a
good predictor of land and atmosphere interactions. Analyzing a shallow soil layer can provide key information
for the detection of soil aridity conditions that are directly related with the loss of soil biodiversity and, therefore,
with soil productivity. Thus, soil moisture at the surface is directly affected by drought conditions and could be
also used as an indicator (i.e., proxy) of the water contained at deeper layers. Since we cannot measure *in situ* soil
moisture at high spatial resolution due to logistical constraints (i.e., because is expensive or time consuming), we
propose the use of multiple satellite remote sensing sensors (e.g., from ESA or NASA) as an alternative to obtain
drought-relevant information on soil moisture at the national scale. The study period (2009 – 2015) was marked
by successive droughts across Brazil, registered and confirmed by different monitoring instruments such as the
Integrated Drought Index (IDI), which combines the Standardized Precipitation Index (SPI) and the Vegetation
Health Index (VHI) (Cunha et al., 2019) and Municipal Emergency Declarations all over the country.
The purpose of this study is showing the advantages and disadvantages of integrating satellite soil moisture
observations into drought monitoring across Brazil on a biome basis. We show the differential impact of drought
on the soil moisture of different biomes at a national scale (using Brazil as a case study).
One main limitation is that satellite measurements of soil moisture provide indirect estimates of soil moisture only
in the topsoil (eg., 0-5cm), and unfortunately do not provide a direct metric of soil water storage. While soil
moisture at the surface is a key indicator of soil and atmosphere interactions, topsoil moisture does not account
entirely for the water used by plants to grow. The capacity of plants to grow can be measured also with satellite
information in the form of primary productivity estimates (Li et al., 2019). Therefore, we also explore the
correspondence between satellite soil moisture and primary productivity trends for each biome in Brazil. Both soil
moisture and vegetation productivity are ecosystem variables directly affected by drought conditions.
Understanding how soil moisture and vegetation productivity on each biome is affected by drought conditions
from different perspectives (in our case superficial soil moisture) is crucial to assess their resilience. It is also
important to provide evidence-based orientations to drought mitigation and soil conservation plans.


**2. Methodology**
**2.1. Study area**
Brazil is the largest country in Latin America with a total area of 8,456,510 km², located between 05º10' N to
33º44' S (IBGE, 2017). The continental dimension of the country implies a complex spatial heterogeneity of
environmental conditions resulting in  six main biomes: Amazon, Atlantic Forest, Caatinga, Cerrado, Pampas and
Pantanal (Fig. 3a).

*The Amazon* biome is mainly characterized by rainforest areas (Overbeck et al. 2015). It represents 49.5% of Brazil's total area, or 4,196,943 km² (IBGE, 2019). It has an equatorial climate, with temperatures between 22°C and 28°C and torrential rains distributed throughout the year. The geomorphology of the Amazon biome is quite diverse, presenting plateaus, plains, and depressions. Soils are generally clayey, iron-rich and with high soil organic carbon content. The Amazon biome is well known for its biodiversity and its large number of rivers and water bodies, which account for the world's greatest surface green water reserves (IBGE 2004).

*The Atlantic Forest* biome covers 13% of the total area of Brazil (1,110,182 km²). It comprises an environmental heterogeneity that incorporates high elevations, valleys, and plains. The Atlantic rainforest occupies the whole continental Atlantic coast of Brazil. This biome has a subtropical climate in the south and a tropical climate in central and northeast portions. The Atlantic rainforest is characterized by heavy rainfall influenced by the proximity of the ocean and winds that blow inward over the continent (IBGE, 2004). Although it is just a small fraction of the size of the Amazon rainforest, the Atlantic Forest still harbors a range of biological diversity comparable to that of the Amazon biome (The Nature Conservancy, 2015), with high soil carbon reserves (Guevara et al., 2018). The Atlantic Forest is recognized as the most degraded biome of Brazil with only 12% of the original biome preserved (SECOM, 2012).

*Caatinga* is the driest biome of Brazil and comprises an area of 844,453 km² stretching over nine federal states and covering nearly 10% of the total area of Brazil (IBGE, 2019). Semiarid climate is predominant across this biome (BSh type) with an average annual rainfall below 800 mm (Alvares et al., 2013), but high temperatures influence high potential evapotranspiration rates that exceed 2,500mm/year (Campos, 2006). Overall, the Caatinga is characterized by reduced water availability and a very limited storage capacity of rivers, which are mainly intermittent, with just a few exceptions that are perennial through streamflow regulating reservoirs during the dry season (CENAD 2014). Caatinga soils are generally shallow (0-50 cm), with a bedrock that is commonly exposed to the surface, limiting water infiltration processes and the recharge of local aquifers (Cirilo, 2008).

*The Cerrado* is the second largest biome of Brazil, characterized by large savannas (Overbeck, et al 2015) covering 2,036,448 km², and representing 23.3% of the country (IBGE, 2019). It extends from the central south of Brazil until the north coastal strip, interposing between the Amazon, Pantanal, Atlantic Forest, and the Caatinga biomes (IBGE, 2004). The dominant climate in the Cerrado is warm tropical sub-humid, with only two distinct seasons, dry winters and wet summers with torrential rains (Overbeck et al. 2015). The annual precipitation in this region varies between 600-2200 mm, where the bordering areas with the Caatinga are the driest and the bordering areas with the Amazon rainforest the wettest. Soils are diverse and include a variety of dystrophic (low inherent fertility and/or strongly weathered profile), acidic, and aluminum-rich conditions. Currently, the Cerrado hosts the largest rural expansion in Brazil, resulting in environmental degradation, biodiversity loss, and soil erosion and limited water availability. It is classified as the most endangered savannah on the planet and one of the 34 global hotspots (Ioris, Irigaray and Girard 2014).

*The Pampas* biome is located at the extreme south of Brazil and covers 2.1% of Brazil's total area (176,496 km²). It is mainly characterized by grasslands and shrublands (Overbeck et al. 2015). The region has a wet subtropical climate, characterized by a rainy climate throughout the whole year, with hot summers and cold winters, where temperatures fall below freezing (IBGE 2004). The Pampas comprises an environmental set of different lithology types and productive soils (e.g., carbon-rich), mainly under flat and smooth undulating terrain surfaces.

Pantanal is the biome with the smallest territorial extension of Brazil, covering 1.8% (150.355 km²) of the
country's total area (IBGE, 2004). It is located at the left margin of the Paraguay River and shared by Brazil,
Bolivia and Paraguay.
*The Pantanal* is by a vast extent of poorly drained lowlands that experiences annual flooding from summer to fall
months (January–May) (Assine and Soares, 2004). The climate of the Pantanal is hot and humid during the
summer and cold and dry in winter (Ioris, Irigaray and Girard 2014). Precipitation varies from 1000-1400 mm per
year, and rains are predominant from November to April. Average annual temperature is 32°C, but the dry season
(May to October) has an average temperature of 21°C and it is not uncommon to have >100 days without rain
(Ioris, Irigaray and Girard 2014). In the last two decades, temperature in the Pantanal has consistently risen and
more humid than normal events as well as dryer than normal events have both increased (Marengo et al 2010).

**2.3. Environmental variability of Brazilian Biomes**
We used 1x1 km environmental gridded data to characterize the environment variability of the biomes. Data was
provided by worldgrids.org, an initiative of ISRIC – World Soil Information Institute. This dataset compiled
information from: 1) digital terrain analysis to represent topographic gradients, 2) gridded climatology products
(e.g., precipitation and temperature), 3) remote sensing imagery, to represent land cover and vegetation spatial
variability, and 4) legacy soil or rock type maps. We used 110 layers derived from this dataset. A list of all
available information contained in the worldgrids.org project is available at Reuter & Hengl (2012). We used
multivariate statistics in the form of principal component analysis (PCA) to linearly decompose the worldgrids.org
dataset and identify relationships among the major environmental characteristics of Brazilian biomes. PCA is an
analysis where a group of potentially correlated variables are decomposed in orthogonal space and therefore
uncorrelated principal components. PCA analysis is useful to reduce data dimensionality to avoid the potential
effects of statistical redundancy (multicollinearity) in further interpretations. Here, we use the PCA as an
exploratory technique to visualize/characterize/interpret the environmental variability of Brazil's biome and
assume that environmental differences in the biomes could support the hypothesis of different soil moisture
response to drought.

**2.4. Municipal emergency declarations due to drought across Brazil**
Municipal Emergency Declarations (MEDs) are administrative tools to inform the federal government that the
magnitude of the disaster has surpassed local public capacities to respond and manage the installed crisis. The
recognition of MEDs by the federal government is based on field visits (when possible) and technical analysis of
social, economic and climatological data that can support the petition. In the case of drought, data analysis is
generally based on, but not limited to, private agricultural losses, level of local reservoirs, and precipitation data
combined. Once the federal government recognizes that there is indeed a disaster, it establishes a legal situation
where federal funds can be used to assist the affected population and recover essential services disrupted by the
disaster (National Secretary of Civil Defense and Protection of Brazil 2017).
To determine drought distribution across the six Brazilian biomes, we retrieved official MEDs due to drought in
Brazil from 2009 to 2015. This information is public and can be accessed on the website of the Ministry of
National Integration of Brazil. First, we downloaded the historical series of MEDs in Brazil from 2009 to 2015.
Then, we isolated the municipalities who declared emergency or public calamity due to drought from all other
disasters. The last step was to cross this data with the boundaries of the six Brazilian biomes and discover the
intensity and distribution of drought in each biome during the study period.
**2.5. Soil Moisture and Primary Productivity Trends across Brazil**
To analyze soil moisture trends during a period of successive droughts (2009-2015) across Brazilian biomes, we
acquired remotely sensed soil moisture information from the European Space Agency (Liu et al. 2011). This soil
moisture product has a daily temporal coverage from 1978 to 2016 and a spatial resolution of 0.25 degrees (~27x27
km grids). To represent vegetation primary productivity we use estimates from the OCO-2-based SIF product
(GOSIF) and linear relationships between SIF (Solar-induced chlorophyll fluorescence) and GPP (gross primary
production) used by Li and Xiao (2019) to map GPP globally at a 0.05∘ spatial resolution and 8-day time step. We
calculated monthly averages from soil moisture and primary vegetation datasets for further statistical analysis
using only information between 2009 and 2015. All available information was harmonized into a geographical
information system using the same projection system and spatial integrity.
**2.6. Data Analysis**
We based our statistical analysis in a regression matrix containing 10,000 representative random spatial locations
(e.g., latitude and longitude) across the biomes of Brazil (Fig. 3b) which were selected using standard re-sampling
techniques (i.e., bootstrapping). Over 30% of the area for every biome is represented in the random selection. We
randomize our statistical sampling with the ultimate goal of maximizing the accuracy of the results. We used a
representative sample for improving the visualization of points cloud and a better understanding of differences on
the five biomes in the statistical multivariate space. Finally, we extracted to these random points the environmental
data and the values of the available satellite soil moisture and primary productivity time series.
To detect trends on satellite soil moisture and primary productivity time series during the study period, we used
median based linear models calculated for each point with available satellite data. These non-parametric analyzes
are known as Theil – Sen regressions (Sen 1968; Theil 1992) with repeated medians (Siegel 1982). This method
uses a robust estimator for each point in time, where the slopes between it and the other points are calculated
(resulting n-1 slopes), and then the median and the significance of the trend are reported.
The satellite soil moisture source has intrinsic quality limitations across areas where vegetation has more water
than soil (McColl et al. 2017), including areas across the lower Amazon watershed, the Pantanal or the Pampas
biomes. For these areas we used the sparse points with available satellite soil moisture information and generated
predictions of soil moisture trends based on geostatistical analyses, such variogram fitting and Ordinary-Kriging
modeling. Ordinary-Kriging assumes that the target variable (soil moisture trends) is controlled by a random field
(main reason why we base our analysis in a random sampling strategy) and that shows a quantifiable level of
spatial structure and autocorrelation (Hiemstra et al. 2009). We performed an automatic variogram analysis to
assess the spatial structure and autocorrelation of satellite soil moisture records. For the variogram analysis we
computed the relationships between the distance of randomly distributed soil moisture observations and the
accumulated variance of their respective values. We used the aforementioned relationships to predict the satellite
soil moisture trend in areas where no data is available and also provided a spatial explicit measure of error
following a geostatistical framework (Hiemstra et al. 2009, Llamas et al., 2020). In contrast, the primary
productivity dataset used here has complete coverage across Brazil. We show both the interpolated maps of soil
moisture trends and the trend map of the primary productivity of vegetation.

**3. Results and Discussion**
**3.1. Drought in Brazil from 2009 to 2015**
This analysis of Municipal Emergency Declarations (MEDs) confirmed that the period from 2009 to 2015 was,
indeed, marked by successive droughts countrywide (Fig. 2). During this period, Brazil had a total of 12,508
declarations of emergency or public calamity due to drought all over its territory (Ministry of National Integration
of Brazil 2018), which affected directly 33 million people and caused economic losses around US$ 6,5 billion
(EM-DAT 2018).
Proportionally, Caatinga is the biome with more MEDs per municipality, followed by the Atlantic Forest, Cerrado,
Pampas and the Amazon respectively (Fig. 2). The only biome with no MEDs due to drought during this period
is the Pantanal, which is a natural wetland that covers only 1.8% of the national territory (Overbeck et al. 2015).

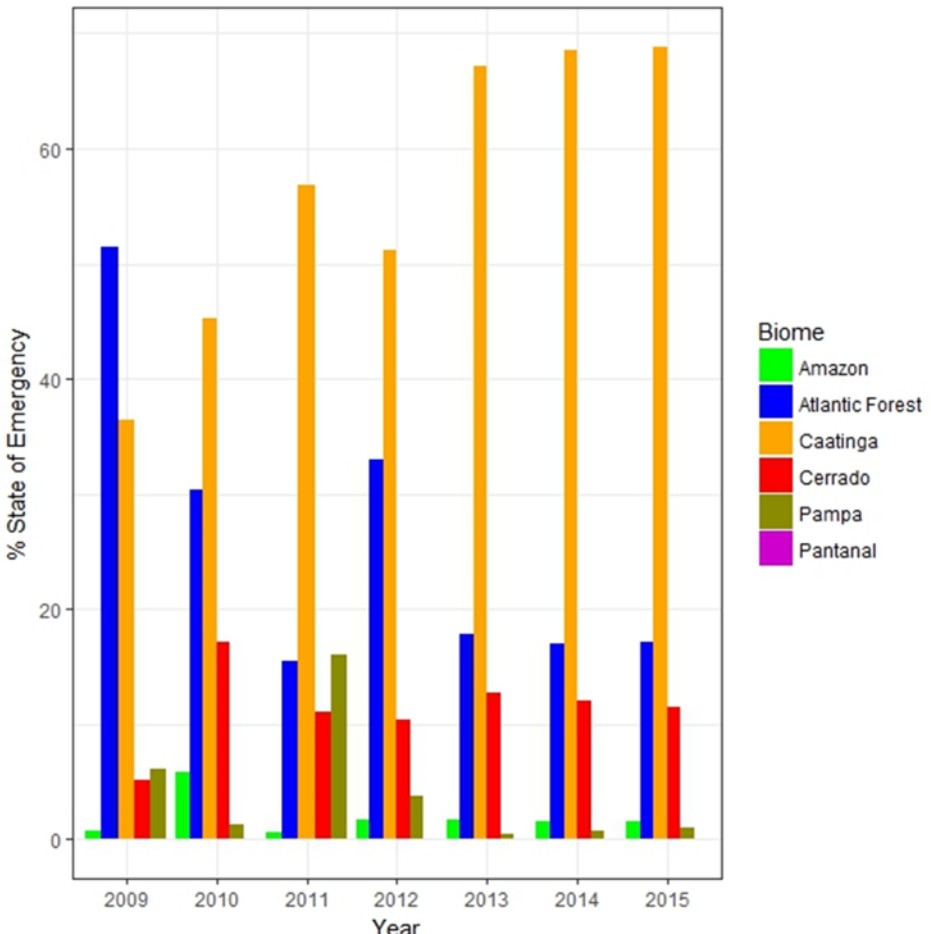


**Figure 2: Percentage of municipalities declaring emergency or public calamity due to drought in Brazil**
**from 2009 to 2015**

When considering climatological data from the Integrated Drought Index (IDI), which combines the Standardized
Precipitation Index (SPI) and the Vegetation Health Index (VHI), Cunha et al. (2019) discovered that since 1962,
when drought events started to be recorded in Brazil, only between 2012 and 2014 droughts occurred concurrently
in the six biomes of the country. The IDI also showed that the hydrological year of 2011/2012 (October 2011 to
September 2012) was the driest of the historical series, except in the South region, where the Pampas biome is
located. During the period of study (2009-2015), the most severe drought events occurred in the northeast region
(where the Caatinga predominates), in the central west region (where the Cerrado predominates), and in the
southeast region (where there is a mix of Cerrado and Atlantic Forest). Even though the climatological data from
the IDI show some inconsistencies with the MEDs per biome, in general terms, it reinforces that the study period
was marked by simultaneous droughts across all biomes of Brazil.

**3.2. Environmental gridded information of Brazilian Biomes**
The environmental characterization of Brazilian biomes showed a clear differentiation of three major groups (Fig.
3a and b). These results support the expectation that drought would have a differential impact on soil moisture
dynamics in each of the six biomes (see section 3.3). This expectation is supported because each biome shows
differences on the spatial configuration of environmental soil moisture drivers, as revealed by the PCA analysis
(Fig. 3b) as described below.
From the 110 environmental layers of information we used to represent the major environmental conditions across
Brazil (see list of available layers in http://worldgrids.org/doku.php), at least 50 principal components were
needed to capture >80% of total variance. The first and second component explained >25% of variability (Fig.
3b) and the variables that represented most of the variance in the first and second components were the digital
elevation model (r=0.5) and the topographic wetness index (r=0.31) respectively. These two variables are directly
related to the spatial variability of soil moisture dynamics as seen in other regional studies (Guevara and Vargas
2019). Across these principal components (i.e., PC1 and PC2), we found a clear separation of three major groups
of data in the statistical space (Fig. 3c). The Amazon biome forms the larger group of values, followed by another
group composed mainly by the Atlantic forest and the Pampas. The Caatinga and Cerrado biomes form a third
larger group and the remaining Pantanal show a close but independent variability (Fig. 3c). These groups are
located on different quadrants of the plane between the first two PCs (Fig. 3c). Thus, these differences could
influence soil moisture response in these major groups at the biome level.

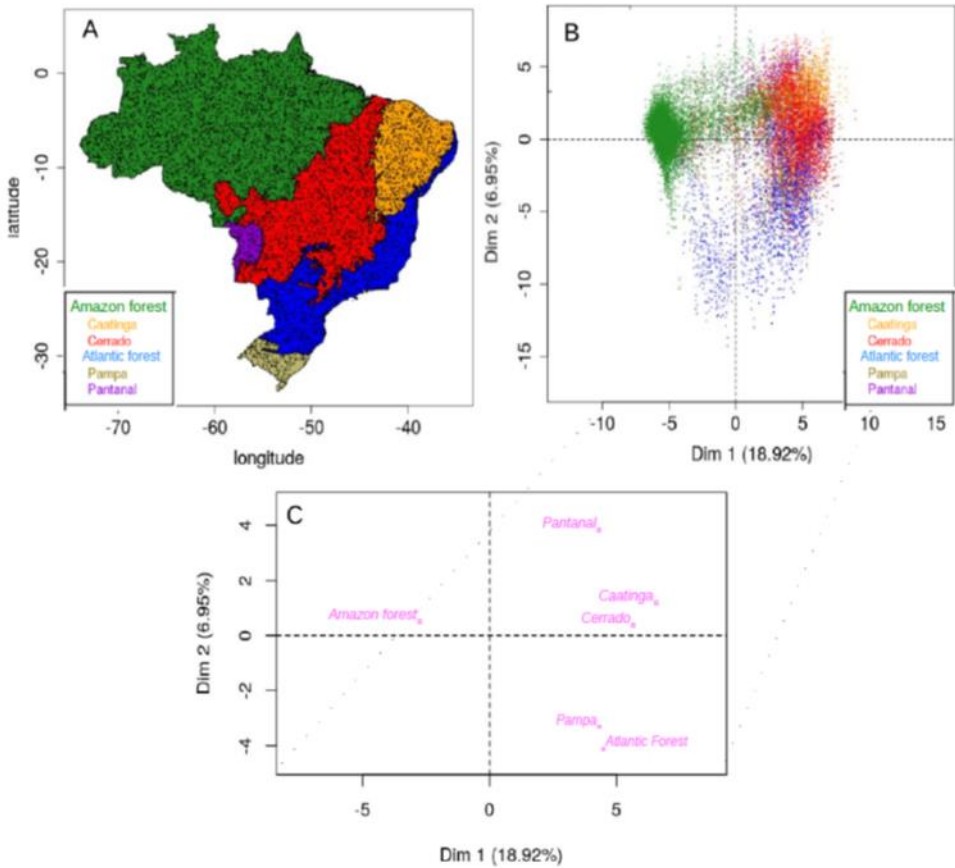


**Figure 3: (a) The six biomes of Brazil. (b) Plane of the first and second PCAs showing the orthogonal and environmental variability of Brazil's biomes and (c) Clustering results showing the main values of each biome dataset and their proximity across the planet between PCAs one and two.**


**3.3. Drought assessment: Soil Moisture Trends Across Brazilian Biomes**

Our analysis of satellite soil moisture at national level showed a soil moisture decline of -0.5% per year (p<0.1) in Brazil from 2009 to 2015 (Fig. 4).

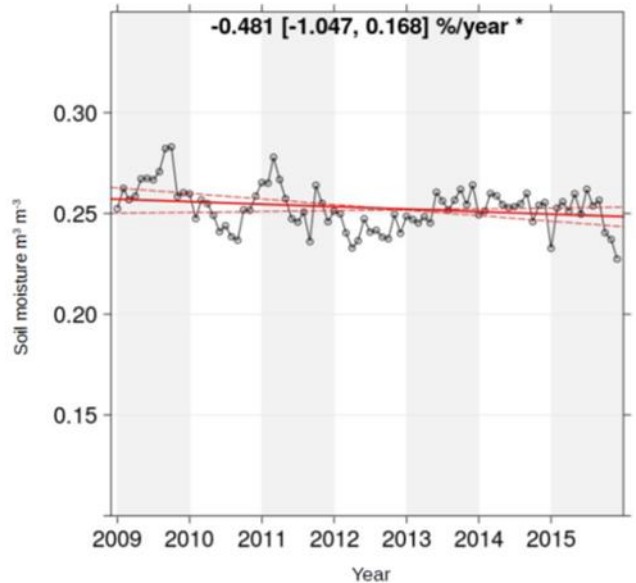

**Figure 4: Brazil soil moisture trend from 2009 to 2015**




When considering variations of soil moisture per biome, our data suggests that the largest soil moisture decline in
Brazil was found in the Caatinga biome with a persistent negative trend (-4.4% in soil moisture per year (p<0.001))
from 2009 to 2015 (Fig. 5a). In contrast, Amazon, Cerrado and Atlantic Forest biomes showed no significant trend
on soil moisture. Pampas and Pantanal biomes showed a significant increase in soil moisture of 1.6% and 4.3%
respectively per year (p<0.001) during the same period (Fig. 5e and f). Thus, the combination of environmental
variables and satellite soil moisture records was able to identify drought dominated areas such as Caatinga and
Cerrado from water-surplus dominated areas, such as Pantanal and Pampas. These results are also useful to prevent
agricultural risk from water failure (decline or surplus) and monitor important ecosystem services of large and
more inaccessible areas such as the Amazon forest and the Cerrado (Fig. 3).






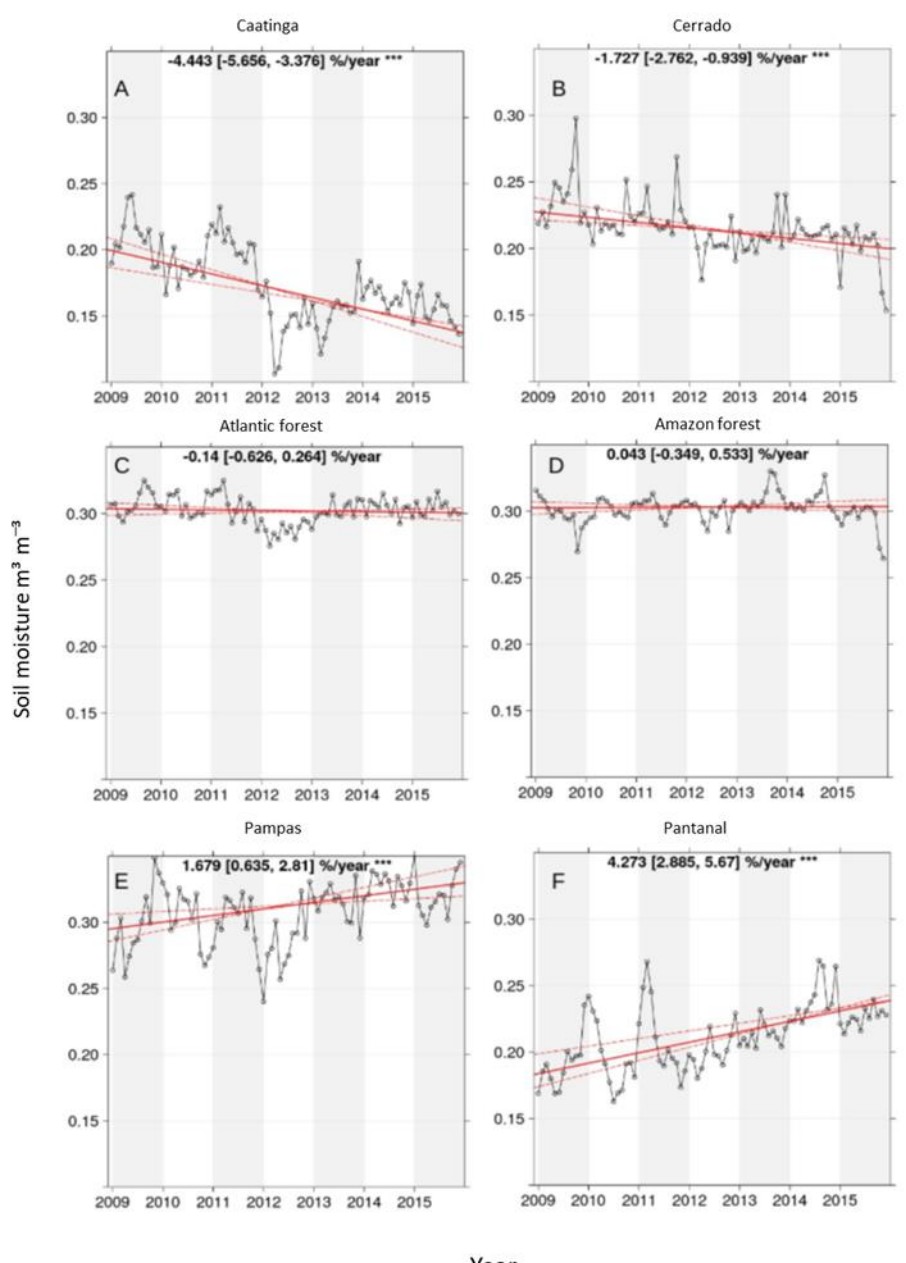

**Figure 5: Soil moisture trends across Brazil. (a) Caatinga (n=921), (b) Cerrado (n=2410), (c) Atlantic Forest (n=1394), (d) Amazon (n=4819), (e) Pampas (n=231), and (f) Pantanal (n=179). The values in every graph show the slope percentages of changes. Red solid line showed the mean trend and red dashed lines show the standard deviation trend. \*\*\* (p<0.01)**

A closer analysis of satellite soil moisture trend in the Caatinga biome shows that this biome did not fully recover from an accentuated soil moisture decrease in 2012 (Fig. 5a). After 2012, there was a slight recovery of soil moisture in 2013, yet a negative trend remains in the following years, most likely because the below average annual precipitation from 2013 to 2015 (Cunha et al., 2019) coupled with human activities commonly found within the boundaries of this biome such as deforestation, unsustainable irrigation and water abstraction (Medeiros 2012;

Travassos and De Souza, 2014). As highlighted by Cunha et al. (2015) intense drought events can reduce the
vegetation resiliency, rendering plants to be more vulnerable to a recurring disturbance. Furthermore, the
vegetation can be durably affected by a drought, if the drought is preceded by another dry year that could
substantially reduce gross primary productivity and other ecosystem processes (Vargas, 2012).
Consistent with previous studies (Zeri et al. 2018) precipitation data indicates that the years 2011, 2012, 2014 and
2015 have been drier as compared to the previous decades. Marengo et al. (2017) also confirmed that, from 2012
to 2015, drought affected hundreds of cities and rural areas with devastating impacts on the agricultural production
and water supply. On the human activities side, data from the National Institute of Spatial Research (INPE, 2018)
reveals that 45% of the Caatinga biome is degraded and 7.2% of its soil is already exposed. In addition, the
Caatinga has been exposed to continuous land cover changes and less than 1% of the region is a strictly protected
area (Leal et al., 2005; Morim et al., 2013). Thus, our results: (a) provide insights to identify geographical areas
that could be preserved due to its capacity for providing blue and green water; and (b) could be part of a monitoring
system for optimizing the limited water inputs and supply in this semiarid ecosystem (i.e., for agricultural
planning).
Persistent and prolonged soil moisture decline could also negatively affect Caatinga's biodiversity, one of the
world's plant biodiversity centers (Leal et al. 2005). The vegetation and soils of the Caatinga are exposed to 8-10
dry months per year (Santos et al. 2014), and more than 90% of the Caatinga biome is non-forest vegetation. Just
~20% of the biome has native vegetation, which is better adapted to support drought events and store higher
amounts of water (Santos et al. 2014; Overbeck et al. 2015). Tomasella et al. (2018) using NDVI values for high
density vegetation and bare soil showed that recurrent droughts are accelerating the degradation and desertification
processes in the Caatinga.
The combination of these regional factors together with the effect of teleconnections such as the ENSO (El Nino
Southern Oscillation) and other land atmosphere interactions (Kouadio et al. 2012) make the Caatinga biome in
Brazil the most vulnerable biome to the recurrent droughts and consequently, prolonged soil moisture deficit
condition. (Marengo et al. 2017).
Therefore, we highlight the need to include urgent actions such as reforestation and efficient use of underground
water into drought mitigation plans for this biome to reduce future soil moisture decline. It is noteworthily that
this biome is already presenting agricultural deficits and desertification areas due to natural and anthropogenic
phenomena (Nascimento and Alves 2008; Sheffield and Wood 2008; Medeiros, 2012; Travassos and De Souza
2014). As an example, while studying the desertification process in part of the Caatinga biome, D' Souza,
Fernandes and Barbosa (2008) found high levels of social, economic, and technological vulnerabilities which
could be directly associated with removal of the natural vegetation covering and forest fires for subsistence
agriculture. These human induced changes on soil moisture in the Caatinga are also related with the occurrence
of soil erosion and local desertification processes that influence low agricultural productivity due to diminish soil
moisture and quality of the soil (Nascimento and Alves 2008).
The Atlantic Forest biome didn't show significant positive or negative trends in soil moisture variation during the
studied period. It registered, however, the greatest ups and downs in soil moisture from 2009 to 2015, with high
peaks (2009, 2011 and 2013) followed by abrupt declines in a relatively short time period. After the most intense
period of soil moisture decline in the Atlantic Forest (2009-2012), this biome quickly bounced back to previous
levels of soil moisture, showing capacity to recover from intense soil moisture losses in less than 12 months.

The Amazon biome showed no significant trend of satellite soil moisture data during the analyzed period (Fig. 4d), probably due to data limitations (i.e., data gaps) associated with lack of satellite-derived information (see Methods section). Field-based evidence collected by Anderson et al. (2018) showed a wide range of impacts of drought on the Amazon forest structure and functioning (e.g.: widespread tree mortality and increased susceptibility to wildfires) in 2016 after the 2015 drought, which affected approximately 46% of the Brazilian Amazon biome. However, considering the size and differences in topography in the Amazon biome, the eastern and western areas of the Amazon rainforest may respond differently to drought due to differences in climate conditions and therefore, different sensibility to soil moisture decline. The western portion of the Amazon biome shows higher soil moisture values (and potentially positive soil moisture trends) than the eastern region (Fig. 6a and b). This result is consistent with previous findings describing differences in drought response from east and west portions of this biome (Duffy et al. 2015), suggesting that soil moisture conservation plans and drought mitigation strategies in the Amazon biome should consider the heterogeneity of the region and the different soil moisture feedback from the east and west portions of this biome.

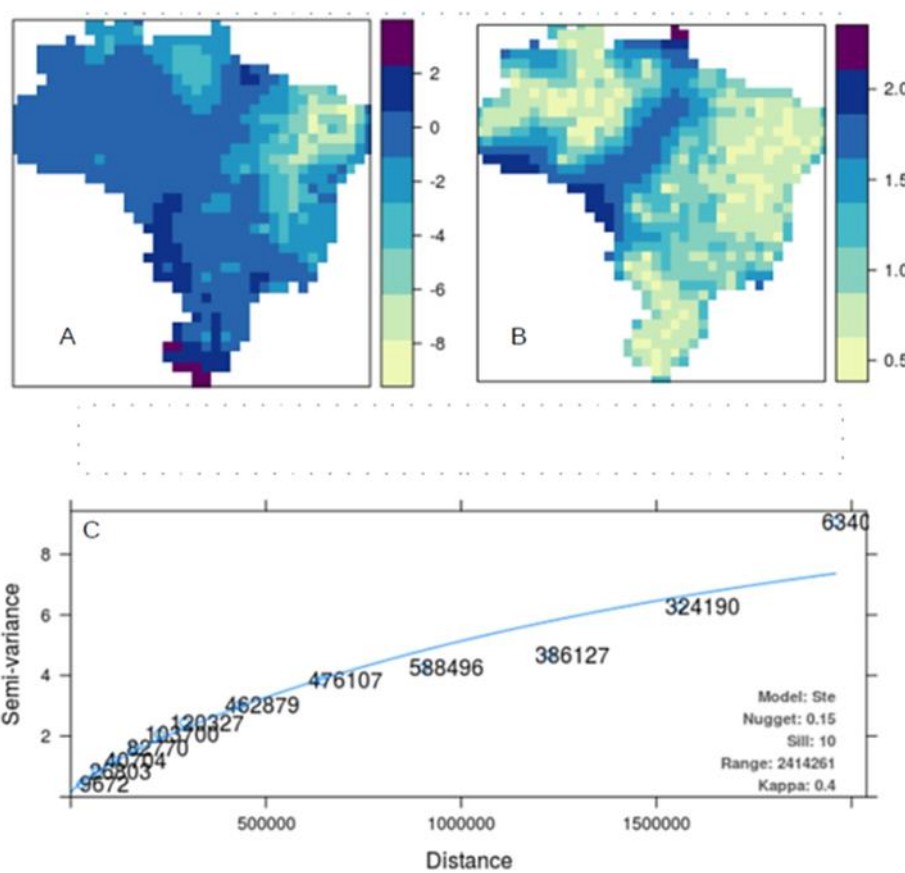

**Figure 6: Geostatistical analysis (Ordinary-Kriging with automatic variogram fitting) of satellite soil moisture across Brazil from 2009 to 2015. (a) The trend prediction of soil moisture 2009-2015. (b) The kriging variance (error map), (c) Variogram fitting parameters and spatial autocorrelation model (blue line) supporting the soil moisture prediction. The numbers around the blue line are the pairs of points available for the interpolation at a specific distance (x-axis)**

The Pampas biome showed a positive trend of ~1.6% per year (p<0.001) during the analyzed period (Fig. 5e), but
with three distinct periods. The year 2009 registered a recovery period of positive soil moisture trend followed by
a steady soil moisture decline until its lowest point in the beginning of 2012. Then, this biome started a consistent
recovery process surpassing previous values of soil moisture trend registered before 2013, showing great capacity
to recover soil moisture after periods of drought. Cunha et al. (2019) showed that in 2012 most of the south region
of Brazil presented drought conditions over an extensive area, with the highest intensity recorded in August 2012.
This intense drought affected the water supply in the rural properties and the agricultural and livestock production.
Even though the Pampas has more than 60% of its biome degraded, especially for cattle raising (Santos and Silva
2012), our data shows that it is gradually increasing soil moisture even during a period of successive droughts
across Brazil. Literature on soil moisture of the Pampas biome characterize this biome as highly vulnerable to
water and wind erosion (Roesch et al. 2009), making it susceptible to soil moisture decline (Duffy et al. 2015).
On the other hand, extended flat landscapes, like the Pampas, show low lateral water transport as a result of low
surface runoff and slow groundwater fluxes, making this biome more suitable to accumulate surface water for
long periods of time (Kuppel et al. 2015).
The Pantanal biome also showed a positive soil moisture trend of 4.3% per year (p<0.001) from 2009 to 2015, the
highest positive trend among all biomes. From 2009 to 2011, there were two extreme events characterized by
sudden soil moisture increase immediately followed by abrupt soil moisture declines. After these two extreme
events, a more stable and consistent positive soil moisture trend was registered from 2011 to 2014. Even though
there was a subtle decline in the soil moisture by the end of 2014, this biome kept an overall positive trend during

388    2015.

The Pantanal and the Pampas biomes are both sub-humid aeolian plains, which make them more susceptible to
experience flood events covering a significant fraction of the landscape for months or even years (Kuppel et al.
2015). Even though our data seems congruent with inundations registered in Pantanal in the beginning of 2011,
when soil moisture trend reached its highest point for the Pantanal biome during the studied period, it did not
capture a reduction of 81% of the total flooded area for the Pantanal biome in 2012, when there was a reduction
of 18% in annual precipitation (Moraes, Pereira and Cardozo 2013). In contrast, our data showed a consistent
positive trend throughout 2012, even though all months of the wet season in 2012 had a decrease in precipitation
ranging from -28.6% in the beginning towards -12.1% in the end of the wet season (Moraes et al. 2013). These
results suggest that, although the analyzed period is characterized by a sequence of dry spells across Brazil
(Marengo et al. 2017), some areas such as the Pantanal region, were able to accumulate soil moisture during that
time.
Detecting an increase in soil moisture does not mean that these biomes should receive less attention to drought
and soil conservation plans. From 2009 to 2015, the Pampas had always a representative municipality declaring
emergency due to drought and has constantly reported economic losses in the agricultural sector. The Pantanal,
during the same period, was not directly impacted by drought at the municipal level, but the highly positive soil
moisture trend deserves further understanding on how it impacts the local ecosystem, as well as agricultural
practices and cattle raising with the ultimate goal to improve food security across Brazil.
Our results support our main hypothesis as we have found evidence that each of the six Brazilian biomes registered
different soil moisture feedbacks to drought during the analyzed period (2009-2015). In practical terms, it means
that drought response and mitigation plans, as well as soil conservation strategies should consider both differences
among and within each biome of Brazil and concentrate efforts and resources to preserve or recover the regions
with greater susceptibility to lose soil moisture during periods of drought. Confirming the value of satellite soil
moisture signals monitoring drought related patterns, we observe the similar trends of soil moisture and the
primary productivity of vegetation across Brazil.

**3.4. Primary productivity trends across Brazil**
We confirm the consistency of our results comparing trends of satellite soil moisture with trends calculated using
the primary productivity (or GPP) datasets. Our results show that all biomes experienced positive and negative
trends of vegetation productivity between the analyzed period of time (Fig. 7). We observe that the major surface
of negative trends of primary productivity of vegetation is across the Caatinga biome and its intersection with
both the Cerrado and Atlantic Forest biomes. Pampas and Pantanal are the biomes with higher surface of positive
primary productivity trends (Fig. 7).

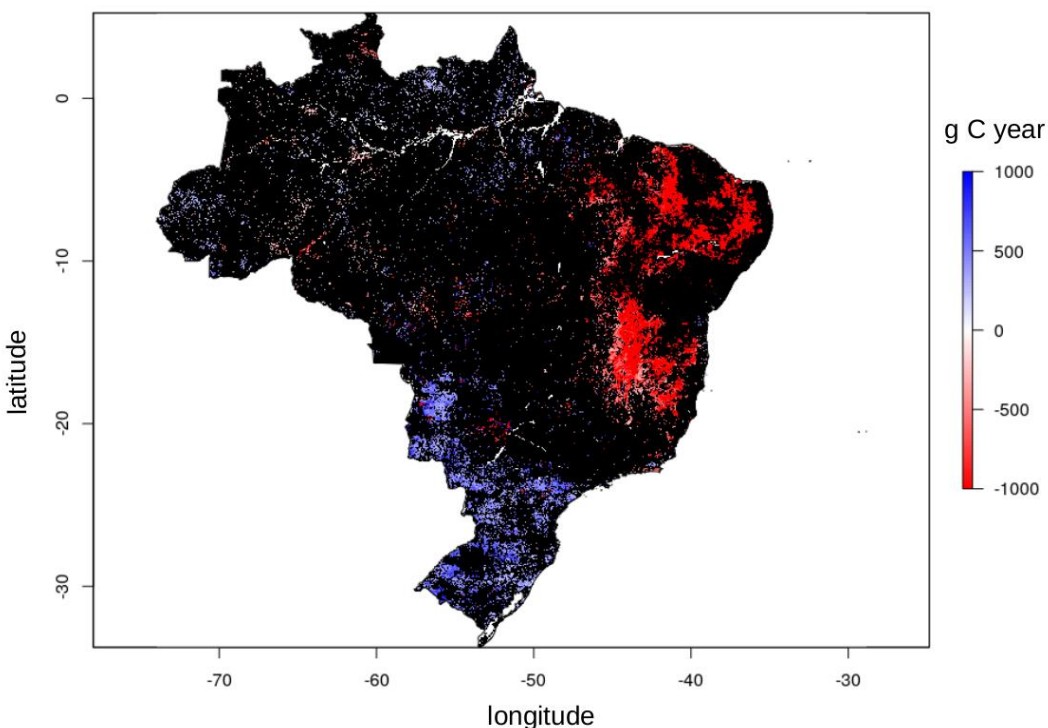

**Fig. 7. Trends of primary productivity of vegetation based on the GOSIF dataset between 2009-2015. Areas**
**in black showed non-significant.**

These results are consistent with the soil moisture trends described on each biome (Fig. 5). Caatinga is the biome
with highest soil moisture decline and highest primary productivity decline. Cerrado and the Atlantic forest are
biomes also experiencing decline in soil moisture and primary productivity. In contrast, the Pampa and Pantanal
experienced an increase in soil moisture levels and increase in primary productivity rates (Fig. 8). Changes in
primary productivity across the Amazon forest were less evident or not significant.  Our results support the use of
satellite soil moisture and primary productivity trends as accurate indicators of drought conditions across Brazilian
biomes.

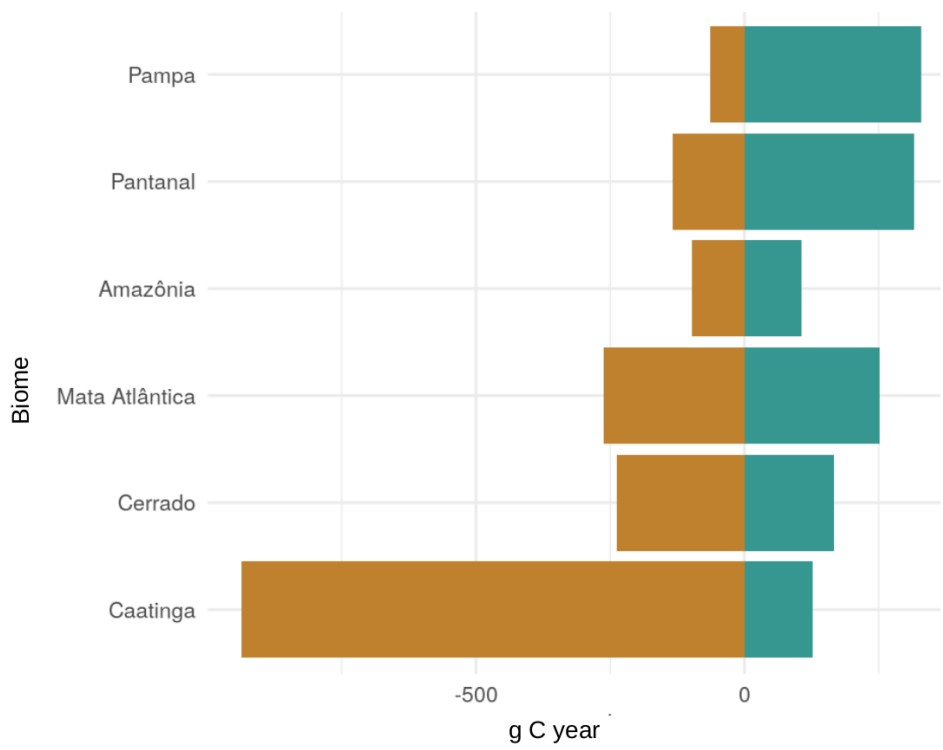



**Fig. 8 Primary productivity trends across Brazilian biomes based in the GOSIF-GPP product across the**

**analyzed period of time (2009-2015).**


**4. Conclusion**

The results of this research reveal an important environmental vulnerability to drought across Brazil. From 2009 to 2015, there was a national decline of soil moisture with a rate of 0.5% year-1. Among all six biomes, Caatinga presented the most severe soil moisture decline (-4.4% year-1), suggesting a need for immediate local soil and water conservation activities. The Atlantic Forest and Cerrado biomes showed no significant soil moisture trends but should be closely monitored for its importance to national food and water security and environmental balance. The Amazon biome also showed no soil moisture trend but a sharp reduction of soil moisture from 2013 to 2015. It is noteworthy that soil moisture from eastern and western portions of the Amazon biome may respond differently to drought. The western portion of the Amazon biome shows potentially more positive soil moisture trends than the eastern region. In contrast, the Pampas and the Pantanal biomes presented a positive soil moisture trend (1.6 and 4.3 % year-1, respectively), which should also be constantly monitored considering the susceptibility of these biomes to floods.

These results are consistent with primary productivity trends (Fig. 8), supporting the effectiveness of satellite soil moisture data to monitor drought impacts at a biome level. This study provides insights about the potential benefits of integrating satellite soil moisture data into drought monitoring and early warning systems and soil conservation plans at national and local levels.




Acknowledgments
FR acknowledges individual scholarship support from CNPq, Science without Boarders program, Brazilian
Federal Government. MG and AVL acknowledge individual fellowship support from CONACyT. RV
acknowledges support from the National Science Foundation CIF21 DIBBs (Grant #1724843).

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
