# Peer review of "The Impact of Drought on Soil Moisture Trends across"

_Natural Hazards and Earth System Sciences, 2020_

## Referee Comment (RC1) · Anonymous Referee #1 · 3 Jul 2020

The authors analyze large scale soil moisture trends, obtained from satellite soil moisture (ESA), as a means to estimate drought risks for specific biomes. They focus on Brazil as case study. Given the variety of biomes and the recurrent droughts in the past decade, Brazil is ideal for studying the impact of droughts. The authors observe different soil moisture trends between biomes, which they attribute generically to their different response and vulnerability to droughts. The authors conclude by proposing the integration of satellite soil moisture observations into drought monitoring.

The paper is well-written and easy to follow, especially given the simple structure "Methodology – Results and discussion – Conclusion". I do have, however, two major comments.

Satellites provide soil moisture for the first 5 cm of soil. This is an important limitation.

[Figure]

The first 5 cm of soil is not representative of soil water storage. For example, in the Amazon or Cerrado soils can be very deep and clayey. Although in tropical soils the organic layer can be thin, deeper layers still represent an important water storage. Also, the rooting system grows below the first 5 cm, so these soil moisture estimates cannot directly inform us on plant available water storage and, consequently, on risks for vegetation productivity. Another important aspect of surface soil moisture is that the first 5 cm are expected to be very dynamic, since it is the first layer exposed to the atmosphere. As a result, one expects a weak autocorrelation, so that it is difficult and ambitious to link the observed soil moisture trends (over 7 years) to the occurrence of specific droughts during the study period. The authors should at least amply discuss this, because this is an important aspect limiting the use of satellite soil moisture.

While reading the manuscript, I was expecting to see the temporal evolution of drought indexes across Brazil. In my opinion, to put their study into context it is key that the authors show how commonly use droughts indexes vary during the study period. One example is the popular Palmer drought index, but there are others. A comparison between these indexes and the soil moisture trends analyzed by the authors might help understand if some information that is missing in drought indexes can be retrieved from soil moisture analyses. In my opinion, this would be critical to evaluate the impact of the paper.

Minor comments

In the introduction the authors say: "Soil moisture decline reduces biomass...". I would be careful here. A soil moisture decline may limit vegetation growth and microbial activity, but only if soil moisture declines below critical water stress thresholds.

In the same paragraph, the sentence "Indeed, temporal variability of soil moisture in a given biome is needed.." is not very clear. I suggest rephrasing and elaborating a bit.

Two paragraphs later "Most work has been focused on the semiarid..". It is not clear why this paragraph is placed here. What work are they referring to? Besides the

abstract, this is the first time they mention Brazil in the paper and the reader does not know why.. This paragraph should probably be moved to later in the introduction.

With a more thorough and quantitative analysis of droughts in Brazil (by means of drought indexes), the discussion should be revisited.

―――――――――――――――

---

## Author Comment (AC1) · 15 Jul 2020

Dear Reviewer,

Thank you for your comments and suggestions.

I am sorry for the late response. We were having some very interesting discussions on how to answer each of your comments.

Please, find the answers below:

Answers

Major Comment #1:

[Figure]

Reviewer comment:The first 5 cm of soil is not representative of soil water storage.

Response: We agree that satellite soil moisture represents only a shallow soil layer and we do not attempt to represent the impact of drought on the complete soil profile based only on satellite soil moisture data. However, the negative impacts of drought can be detected in different components of the water cycle, including superficial soil moisture. The propose of this article is to demonstrate the benefits of integrating satellite soil moisture observations into drought monitoring such as feasibility of soil moisture products, greater spatial coverage (much better than weather stations), high temporal resolution, comparability with other areas of the world, repeatability of results, etc. Having said that, we will edit the document clarifying that we do not mean to represent soil water storage along the complete soil profile.

Reviewer comment: Amazon or Cerrado soils can be very deep and clayey. Although in tropical soils the organic layer can be thin, deeper layers still represent an important water storage. Also, the rooting system grows below the first 5 cm, so these soil moisture estimates cannot directly inform us on plant available water storage and, consequently, on risks for vegetation productivity.

Response: Agree. On the other hand, soil moisture at the first 5cm is a good predictor of land and atmosphere interactions and key for the detection of soil aridity conditions that are directly related with the loss of soil biodiversity and therefore with soil productivity. Thus, soil moisture at the surface is directly affected by drought conditions and could be also used as indicator (i.e., proxy) of the water contained at deeper layers. We will edit the document to highlight the value of superficial soil moisture (0-5cm) as an indicator of drought negative effects. We support these findings with previous literature reporting drought conditions in biomes where we detect soil moisture decline.

Reviewer comment: Another important aspect of surface soil moisture is that the first 5 cm are expected to be very dynamic, since it is the first layer exposed to the atmosphere. As a result, one expects a weak autocorrelation, so that it is difficult and

ambitious to link the observed soil moisture trends (over 7 years) to the occurrence of specific droughts during the study period. The authors should at least amply discuss this, because this is an important aspect limiting the use of satellite soil moisture.

Response: The study period (2009 – 2015) was marked by successive droughts across Brazil, registered and confirmed by different monitoring instruments such as the Integrated Drought Index (IDI), which combines the Standardized Precipitation Index (SPI) and the Vegetation Health Index (VHI) (Cunha et al., 2019) and Municipal Emergency Declarations all over the country. While we agree with some of the limitations of satellite soil moisture, we also highlight that larger drought affected areas (e.g., the Caatinga biome) are consistent with our results. Following the reviewer recommendation, we will increase the discussion about the limiting factors of soil moisture as indicators of drought negative effects (e.g., soil moisture decline).

Major comment #2: Reviwer comment: In my opinion, to put their study into context it is key that the authors show how commonly use droughts indexes vary during the study period. One example is the popular Palmer drought index, but there are others. A comparison between these indexes and the soil moisture trends analyzed by the authors might help understand if some information that is missing in drought indexes can be retrieved from soil moisture analyses. In my opinion, this would be critical to evaluate the impact of the paper.

Response: That's a good suggestion and would help clarify the first major comment and confirm the consistency of our results. For that we will creating a new sub section with a validation of the 5cm SM compared with OCO-2-based SIF product (GOSIF) and linear relationships between SIF (Solar-induced chlorophyll fluorescence) and GPP (gross primary production) used to map GPP globally at a $0.05°$ spatial resolution and 8-day time step, as proposed by Li, X.*, Xiao, J. (2019): https://www.mdpi.com/2072-4292/11/21/2563 We even produced a map of primary productivity trends 2009-2015 for Brazil retrieving data form Orbiting Carbon Observatory-2 (OCO-2) which only confirms our results (attached).

Minor comments

Reviwer comment: In the introduction the authors say: "Soil moisture decline reduces biomass: : :". I would be careful here. A soil moisture decline may limit vegetation growth and microbial activity, but only if soil moisture declines below critical water stress thresholds.

Response: Sure. We will rephrase it.

Reviwer comment: In the same paragraph, the sentence "Indeed, temporal variability of soil moisture in a given biome is needed.." is not very clear. I suggest rephrasing and elaborating a bit.

Response: This sentence is based on the evidences that soil moisture is an integral component of the interactions between climate and the earth's surface that lead to geographical variability of climate. What we are trying to stress here is that beyond being just a resultant statistic, soil moisture itself is an active variable of the local climate and can be add value and precision to the monitoring evaluation of the impacts of drought at the biome level.

Reviwer comment:Two paragraphs later "Most work has been focused on the semi-arid..". It is not clear why this paragraph is placed here. What work are they referring to? Besides the abstract, this is the first time they mention Brazil in the paper and the reader does not know why. This paragraph should probably be moved to later in the introduction.

Response: You're right. We will move the following paragraph up ("In this study, we use satellite soil moisture data from the European Space Agency (ESA) to analyze the impact of drought across all Brazilian biomes...") as a bridge to indroduce the importance of assessing soil moisture to analyze the impact of drought across all Brazilian biomes. The sentence about the semiardi region will be moved further down the indroduction.

Please also note the supplement to this comment:
https://www.nat-hazards-earth-syst-sci-discuss.net/nhess-2020-185/nhess-2020-185-AC1-supplement.pdf

———————————————————

![Map of Brazil showing OCO-2 GPP data for 2009-2015, with a color scale from red (-1000) to blue (1000) g C year]

**Fig. 1.** OCO-2_GPP_BRazil_2009-2015

**Supplement:**

[Figure]

Mapa de tendencias de productividad primaria 2009-2015 para Brazil,

g C year

Fuente: Li, X.*, Xiao, J. (2019) Mapping photosynthesis solely from solar-induced chlorophyll fluorescence: A global, fine-resolution dataset of gross primary production derived from OCO-2. Remote Sensing, 11(21), 2563; https://doi.org/10.3390/rs11212563.

Mapa de tendencias de productividad primaria 2009-2015 para Brazil,

g C year

Fuente de datos: Li, X.*, Xiao, J. (2019) Mapping photosynthesis solely from solar-induced chlorophyll fluorescence: A global, fine-resolution dataset of gross primary production derived from OCO-2. Remote Sensing, 11(21), 2563; https://doi.org/10.3390/rs11212563.

Mapa de tendencias de productividad primaria 2009-2015 para Brazil,

g C year

Fuente: Li, X.*, Xiao, J. (2019) Mapping photosynthesis solely from solar-induced chlorophyll fluorescence: A global, fine-resolution dataset of gross primary production derived from OCO-2. Remote Sensing, 11(21), 2563; https://doi.org/10.3390/rs11212563.

[Figure]

Grafico de tendencias de productividad primaria 2009-2015 para Brazil por bioma,

g C year

---

## Referee Comment (RC2) · Anonymous Referee #2 · 23 Jul 2020

The authors quantify the impact of drought on soil moisture patterns (27 x 27 km grid size) acquired from satellite platforms (European Space Agency, 2009-2015) in six different biomes (Amazon, Atlantic Forest, Caatinga, Cerrado, Pampa, Pantanal) in Brazil. A general decline of -0.5% / year is observed at national level with Caatinga showing the most severe soil moisture decline. The evaluation of this manuscript is based on the following questions: 1) Is it a novel work based on a reliable scientific technique? 2) Is it clearly structured and well-written? 3) Are the experimental design and analysis of data adequate and appropriate to the investigation?

Line numbers are missing, therefore it is difficult to post specific comments. The manuscript is well-written but in its current form seems a good scientific report rather than an article. I struggle to find a novelty in this manuscript since the authors simply

apply known statistical methods to near-surface soil moisture maps. Therefore I highly recommend to re-submit the manuscript by adding something interesting to relate climate drought to soil moisture drought. Some further "quantitative" analysis is required (Van Loon, 2015; von Gunten et al., 2016; Hein et al., 2019; Nasta et al., 2020). 1) Climate drought indexes: please see https://spei.csic.es/home.html and associated references 2) Soil moisture index: please see Hunt et al. (2009), Martínez-Fernández et al. (2015), Sánchez et al. (2016) Satellite measurements provide indirect estimates of soil moisture only in the topsoil, and unfortunately do not provide soil water storage. Moreover dense vegetation cover disturbs the satellite measurements therefore the authors should devote a sub-section on discussing on these issues. Moreover soil moisture observations from 2009 till 2015 do not drive to strong conclusions on temporal evolution, so the authors should warn the reader that this observation is based on very short time series. Usually drought indexes require necessarily at least 30 years of observations. I understand that satellite data provide only short-term temporal evolution but the authors should highlight this issue. Are there any comparisons between satellite-based soil moisture and ground-truthing in Brazil? The authors should also comment on measurement uncertainty

References Hein, A., Condon, L., Maxwell, R. 2019. Evaluating the relative importance of precipitation, temperature and land-cover change in the hydrologic response to extreme meteorological drought conditions over the North American High PlainsHydrol. Earth Syst. Sci., 23, 1931–1950, 2019 Hunt, E.D., K.G. Hubbard, D.A. Wilhite, T.J. Arkebauer, and A.L. Dutcher. 2009. The development and evaluation of a soil moisture index . Int. J. Climatol. 29:747–759. doi:10.1002/joc.1749 Martínez-Fernández, J., González-Zamora, A., Gamuzzio, A. 2015. A soil water based index as a suitable agricultural drought indicator. Journal of Hydrology 522, 265–273 Nasta, P., C. Allocca, R. Deidda, N. Romano. 2020. Assessing the impact of seasonal-rainfall anomalies on catchment-scale water balance components. Hydrol. Earth Syst. Sci. 24:1-17 Sánchez, N., Á. González-Zamora, M. Piles and J. Martínez-Fernández. 2016. A New Soil Moisture Agricultural Drought Index (SMADI) Integrating MODIS and SMOS

Products: A Case of Study over the Iberian Peninsula. Remote Sensing. 8, 287; doi:10.3390/rs8040287 Van Loon, A.F. 2015. Hydrological drought explained. WIREs Water 2015, 2:359–392. doi: 10.1002/wat2.1085 von Gunten, D., T. Wöhling, C. P. Haslauer, D. Merchán, J. Causapé, and O. A. Cirpka. 2016. Using an integrated hydrological model to estimate the usefulness of meteorological drought indices in a changing climate. Hydrol. Earth Syst. Sci., 20, 4159–4175

---

## Author Comment (AC2) · 20 Aug 2020

Dear Reviwer #2,

Thank you so much for all your comments and suggestions. We had great discussions regarding your interaction and I am sure it will result in some improvements on our manuscript. Please, find bellow our answers.

All the best.

————

Reviewer comment: The manuscript is well-written but in its current form seems a good scientific report rather than an article. I struggle to find a novelty in this manuscript

since the authors simply apply known statistical methods to near-surface soil moisture maps. Therefore I highly recommend to re-submit the manuscript by adding something interesting to relate climate drought to soil moisture drought.

Response: There are two main novelties in this article. First, we show the differential impact of drought on the soil moisture of different biomes at a national scale (using Brazil as a case study). For the best of our knowledge, there are no published articles about this issue. Understanding how each biome is affected by drought conditions from different perspectives (in our case superficial soil moisture) is crucial to assess their resilience and provide a more complete evidence-based orientations to drought mitigation and soil conservation plans. Furthermore, this data set has not been used by the disaster management community (our target audience) as a complementary source of knowledge on the systemic impact of drought at national and local scales. Motivated by this knowledge gap and the availability of this dataset on soil moisture, we present some features of using this satellite soil moisture product to drought monitoring against other approaches, for example feasibility of the soil moisture product, high temporal resolution, that the satellite moisture product is done from radar data that in theory should be more useful to explain soil properties, comparability with other areas of the world, repeatability of results, etc.

_____ Reviewer comment: Some further "quantitative" analysis is required (Van Loon, 2015; von Gunten et al., 2016; Hein et al., 2019; Nasta et al., 2020). 1) Climate drought indexes: please see https://spei.csic.es/home.html and associated references 2) Soil moisture index: please see Hunt et al. (2009), Martínez-Fernández et al. (2015), Sánchez et al. (2016) Satellite measurements provide indirect estimates of soil moisture only in the topsoil, and unfortunately do not provide soil water storage.

Response: We don't agree further quantitative analysis is required in terms of a drought analyses. As stated before, this article has the purpose of showing the advantages and disadvantages of integrating satellite soil moisture observations into drought monitoring across Brazil (and other countries) on a biome basis, and not creating another drought

index. It is targeted on disaster management communities across the globe, which still lack information and scientific evidences on how each biome respond to drought conditions especially considering our present climate emergency. However, to confirm the consistency of our results, we will create a new sub section with a validation of the 5cm SM compared with OCO-2-based SIF product (GOSIF) and linear relationships between SIF (Solar-induced chlorophyll fluorescence) and GPP (gross primary production) used to map GPP globally at a $0.05°$ spatial resolution and 8-day time step, as proposed by Li, X.*, Xiao, J. (2019): https://www.mdpi.com/2072-4292/11/21/2563 We even have already produced a map of primary productivity trends 2009-2015 for Brazil retrieving data form Orbiting Carbon Observatory-2 (OCO-2) which only confirms our results.

_____ Reviewer comment: Moreover dense vegetation cover disturbs the satellite measurements therefore the authors should devote a sub-section on discussing on these issues.

Response: We agree in that specific conditions of dense vegetation can affect the quality of soil moisture measurements. We recognize that dense vegetation conditions e.g., across specific areas of tropical rain forest with dense vegetation are mainly located in the Amazon forest. That was the main reason why we selected a coarse scale but ecologically meaningful delineation of Brazilian biomes, to avoid the lack of information across specific areas with dense vegetation. At the biome level, we observe that the Amazon biome has the higher area of dense vegetation (but not all Amazon is dense vegetation), and probably that is the reason why we don't observe significant trends in this biome. Other biomes are not affected by this issue. We contribute with a prediction of soil moisture trends across all the country using a geostatistical approach (Figure 6 of submitted paper) aiming to contribute with better information across areas less represented with the available satellite data. We will clarify the effects of dense vegetation in satellite soil moisture and will highlight the scale of our work. We will also demonstrate in the new version of the paper that at the biome scale the trends of soil

moisture are consistent with trends in vegetation productivity data.

______ Reviewer comment: Moreover soil moisture observations from 2009 till 2015 do not drive to strong conclusions on temporal evolution, so the authors should warn the reader that this observation is based on very short time series. Usually drought indexes require necessarily at least 30 years of observations. I understand that satellite data provide only short-term temporal evolution but the authors should highlight this issue.

Response: You're right. Drought indexes require at 30 years of observation. However, the objective was not to create a new drought index based on satellite soil moisture data, but to show the potential to use this data to have a broader comprehension on the impacts of drought on different ecological systems. The study period (2009 – 2015) was marked by successive droughts across Brazil, registered and confirmed by different monitoring instruments such as the Integrated Drought Index (IDI), which combines the Standardized Precipitation Index (SPI) and the Vegetation Health Index (VHI) (Cunha et al., 2019) and Municipal Emergency Declarations all over the country. Therefore, the period of study, even though short, it is justified because of the widespread drought conditions across the country.

______ Reviewer comment: Are there any comparisons between satellite-based soil moisture and ground-truthing in Brazil? The authors should also comment on measurement uncertainty.

Response: No, as far as we are aware. Measurements of ground soil moisture are recent in Brazil and data availability is still low due to maintenance and spatial coverage. The largest network (Zeri et al. 2018) covers only the Brazilian semiarid region and measurements started in 2016. However, data transmission problems and lack of funding for regular maintenance make it difficult to establish long-term time series, which are essential to robust statistical analysis. Here is where the relevance of high temporal satellite soils moisture relies. We use a soil moisture product that is constantly improved and areas with potentially wrong measurements

are removed by the source. We alternative validate our work comparing our soil moisture trends with the actual emergency declaration calls from all municipalities in each biome (Fig 2). We also observe environmental differences on each biome (Fig 3) suggesting potential differences in soil moisture drivers. We also observe similar trends of soil moisture and vegetation productivity across all biomes (see Figure 1 of previous response https://editor.copernicus.org/index.php/nhess-2020-185-AC1.pdf?_mdl=msover_md&_jrl=7&_lcm=oc108lcm109w&_acm=get_comm_file&_ms=86004&c=185019&salt=18517131 We will comment on the measurement uncertainty and how the scale of biomes is appropriate for our national assessment.

Zeri, M., Alvalá, R.C.S., Carneiro, R., Cunha-Zeri, G., Costa, J.M., Spatafora, L.R., Urbano, D., Vall-Llossera, M., Marengo, J., 2018. Tools for communicating agricultural drought over the Brazilian Semiarid using the soil moisture index. Water 10, 1421. https://doi.org/10.3390/w10101421
* * *

---

## Author Response (AR3)

Major Comment #1:
**Reviewer comment:** The first 5 cm of soil is not representative of soil water storage.
*Response: We agree that satellite soil moisture represents only a shallow soil layer and we do not attempt to represent the impact of drought on the complete soil profile based only on satellite soil moisture data. However, the negative impacts of drought can be detected in different components of the water cycle, including superficial soil moisture. The propose of this article is to demonstrate the benefits of integrating satellite soil moisture observations into drought monitoring and impact evaluation such as feasibility of soil moisture products, greater spatial coverage (much better than weather stations), high temporal resolution, comparability with other areas of the world, repeatability of results, etc.*
*Having said that, we will edit the document clarifying that we do not mean to represent soil water storage along the complete soil profile.*

**Reviewer comment:** Amazon or Cerrado soils can be very deep and clayey. Although in tropical soils the organic layer can be thin, deeper layers still represent an important water storage. Also, the rooting system grows below the first 5 cm, so these soil moisture estimates cannot directly inform us on plant available water storage and, consequently, on risks for vegetation productivity.
*Response: Agree. On the other hand, soil moisture at the first 5cm is a good predictor of land and atmosphere interactions and key for the detection of soil aridity conditions that are directly related with the loss of soil biodiversity and therefore with soil productivity.*
*Thus, soil moisture at the surface is directly affected by drought conditions and could be also used as indicator (i.e., proxy) of the water contained at deeper layers. We will edit the document to highlight the value of superficial soil moisture (0-5cm) as an indicator of drought negative effects. We support these findings with previous literature reporting drought conditions in biomes where we detect soil moisture decline.*

**Reviewer comment:** Another important aspect of surface soil moisture is that the first 5 cm are expected to be very dynamic, since it is the first layer exposed to the atmosphere. As a result, one expects a weak autocorrelation, so that it is difficult and ambitious to link the observed soil moisture trends (over 7 years) to the occurrence of specific droughts during the study period. The authors should at least amply discuss this, because this is an important aspect limiting the use of satellite soil moisture.
*Response: The study period (2009 – 2015) was marked by successive droughts across Brazil, registered and confirmed by different monitoring instruments such as the Integrated Drought Index (IDI), which combines the Standardized Precipitation Index (SPI) and the Vegetation Health Index (VHI) (Cunha et al., 2019) and Municipal Emergency Declarations all over the country. While we agree with some of the limitations of satellite soil moisture, we also highlight that larger drought affected areas (e.g., the Caatinga biome) are consistent with our results. Following the reviewer recommendation, we will increase the discussion about the limiting factors of soil moisture as indicators of drought negative effects (e.g., soil moisture decline).*

**Major comment #2:**
**Reviewer comment**: In my opinion, to put their study into context it is key that the authors show how commonly use droughts indexes vary during the study period. One example is the popular Palmer drought index, but there are others. A comparison between these indexes and the soil moisture trends analyzed by the authors might help understand if some information that is missing in drought indexes can be retrieved from soil moisture analyses. In my opinion, this would be critical to evaluate the impact of the paper.

*Response*: *That's a good suggestion and would help clarify the first major comment and confirm the consistency of our results. For that we will creating a new sub section with a validation of the 5cm SM compared with OCO-2-based SIF product (GOSIF) and linear relationships between SIF (Solar-induced chlorophyll fluorescence) and GPP (gross primary production) used to map GPP globally at a 0.05° spatial resolution and 8-day time step, as proposed by Li, X.\*, Xiao, J. (2019): https://www.mdpi.com/2072-4292/11/21/2563*
*We even produced a map of primary productivity trends 2009-2015 for Brazil retrieving data form Orbiting Carbon Observatory-2 (OCO-2) which only confirms our results.*

**Minor comments**
**Reviewer comment**: In the introduction the authors say: "Soil moisture decline reduces biomass: : :". I would be careful here. A soil moisture decline may limit vegetation growth and microbial activity, but only if soil moisture declines below critical water stress thresholds.
 *Response: Sure. We will rephrase it.*

**Reviewer comment**: In the same paragraph, the sentence "Indeed, temporal variability of soil moisture in a given biome is needed.." is not very clear. I suggest rephrasing and elaborating a bit.
 *Response: This sentence is based on the evidences that soil moisture is an integral component of the interactions between climate and the earth's surface that lead to geographical variability of climate. What we are trying to stress here is that beyond being just a resultant statistic, soil moisture itself is an active variable of the local climate and can be add value and precision to the monitoring evaluation of the impacts of drought at the biome level.*

**Reviewer comment**: Two paragraphs later "Most work has been focused on the semiarid..". It is not clear why this paragraph is placed here. What work are they referring to? Besides the abstract, this is the first time they mention Brazil in the paper and the reader does not know why. This paragraph should probably be moved to later in the introduction.
**Response:** *You're right. We will move the following paragraph up ("In this study, we use satellite soil moisture data from the European Space Agency (ESA) to analyze the impact of drought across all Brazilian biomes...") as a bridge to introduce the importance of assessing soil moisture to analyze the impact of drought across all Brazilian biomes. The sentence about the semiarid region will be moved further down the introduction.*

**Answers to Reviewer #2**

**Reviewer comment:** The manuscript is well-written but in its current form seems a good scientific report rather than an article. I struggle to find a novelty in this manuscript since the authors simply apply known statistical methods to near-surface soil moisture maps. Therefore I highly recommend to re-submit the manuscript by adding something interesting to relate climate drought to soil moisture drought.
**Response**: *There are two main novelties in this article. First, we show the differential impact of drought on the soil moisture of different biomes at a national scale (using Brazil as a case study). For the best of our knowledge, there are no published articles about this issue. Understanding how each biome is affected by drought conditions from different perspectives (in our case superficial soil moisture) is crucial to assess their resilience and provide a more complete evidence-based orientations to drought mitigation and soil conservation plans. Furthermore, this data set has not been used by the disaster management community (our target audience) as a complementary source of knowledge on the systemic impact of drought at national and local scales. Motivated by this knowledge gap and the availability of this dataset on soil moisture, we present some features of using this satellite soil moisture product to drought monitoring against other approaches, for example feasibility of the soil*

*moisture product, high temporal resolution, that the satellite moisture product is done from radar data that in theory should be more useful to explain soil properties, comparability with other areas of the world, repeatability of results, etc.*

**Reviewer comment:** Some further "quantitative" analysis is required (Van Loon, 2015; von Gunten et al., 2016; Hein et al., 2019; Nasta et al., 2020). 1) Climate drought indexes: please see https://spei.csic.es/home.html and associated references 2) Soil moisture index: please see Hunt et al. (2009), Martínez-Fernández et al. (2015), Sánchez et al. (2016) Satellite measurements provide indirect estimates of soil moisture only in the topsoil, and unfortunately do not provide soil water storage.

**Response:** *We don't agree further quantitative analysis is required in terms of a drought analyses. As stated before, this article has the purpose of showing the advantages and disadvantages of integrating satellite soil moisture observations into drought monitoring across Brazil (and other countries) on a biome basis, and not creating another drought index. It is targeted on disaster management communities across the globe, which still lack information and scientific evidences on how each biome respond to drought conditions especially considering our present climate emergency. However, to confirm the consistency of our results, we will create a new sub section with a validation of the 5cm SM compared with OCO-2-based SIF product (GOSIF) and linear relationships between SIF (Solar-induced chlorophyll fluorescence) and GPP (gross primary production) used to map GPP globally at a 0.05° spatial resolution and 8-day time step, as proposed by Li, X.\*, Xiao, J. (2019): https://www.mdpi.com/2072-4292/11/21/2563*
*We even have already produced a map of primary productivity trends 2009-2015 for Brazil retrieving data form Orbiting Carbon Observatory-2 (OCO-2) which only confirms our results.*

**Reviewer comment:** Moreover, dense vegetation cover disturbs the satellite measurements therefore the authors should devote a sub-section on discussing on these issues.

**Response:** *We agree in that specific conditions of dense vegetation can affect the quality of soil moisture measurements. We recognize that dense vegetation conditions e.g., across specific areas of tropical rain forest with dense vegetation are mainly located in the Amazon forest. That was the main reason why we selected a coarse scale but ecologically meaningful delineation of Brazilian biomes, to avoid the lack of information across specific areas with dense vegetation. At the biome level, we observe that the Amazon biome has the higher area of dense vegetation (but not all Amazon is dense vegetation), and probably that is the reason why we don't observe significant trends in this biome. Other biomes are not affected by this issue. We contribute with a prediction of soil moisture trends across all the country using a geostatistical approach (Figure 6 of submitted paper) aiming to contribute with better information across areas less represented with the available satellite data.*
*We will clarify the effects of dense vegetation in satellite soil moisture and will highlight the scale of our work. We will also demonstrate in the new version of the paper that at the biome scale the trends of soil moisture are consistent with trends in vegetation productivity data.*

**Reviewer comment:** Moreover, soil moisture observations from 2009 till 2015 do not drive to strong conclusions on temporal evolution, so the authors should warn the reader that this observation is based on very short time series. Usually drought indexes require necessarily at least 30 years of observations. I understand that satellite data provide only short-term temporal evolution but the authors should highlight this issue.

**Response**: *You're right. Drought indexes require at 30 years of observation. However, the objective was not to create a new drought index based on satellite soil moisture data, but to show the potential to use this data to have a broader comprehension on the impacts of drought on different ecological systems. The study period (2009 – 2015) was marked by successive droughts across Brazil, registered and confirmed by different monitoring instruments such as the Integrated Drought Index (IDI), which combines the Standardized Precipitation Index (SPI) and the Vegetation Health Index (VHI) (Cunha et*

*al., 2019) and Municipal Emergency Declarations all over the country. Therefore, the period of study, even though short, it is justified because of the widespread drought conditions across the country.*

**Reviewer comment:** Are there any comparisons between satellite-based soil moisture and ground-truthing in Brazil? The authors should also comment on measurement uncertainty.

**Response:** *No, as far as we are aware. Measurements of ground soil moisture are recent in Brazil and data availability is still low due to maintenance and spatial coverage. The largest network (Zeri et al. 2018) covers only the Brazilian semiarid region and measurements started in 2016. However, data transmission problems and lack of funding for regular maintenance make it difficult to establish long-term time series, which are essential to robust statistical analysis. Here is where the relevance of high temporal satellite soils moisture relies.*

*We use a soil moisture product that is constantly improved and areas with potentially wrong measurements are removed by the source. We alternative validate our work comparing our soil moisture trends with the actual emergency declaration calls from all municipalities in each biome (Fig 2). We also observe environmental differences on each biome (Fig 3) suggesting potential differences in soil moisture drivers. We also observe similar trends of soil moisture and vegetation productivity across all biomes (see Figure 1 of previous response https://editor.copernicus.org/index.php/nhess-2020-185-AC1.pdf?_mdl=msover_md&_jrl=7&_lcm=oc108lcm109w&_acm=get_comm_file&_ms=86004&c=185019&salt=18517131109310738). We will comment on the measurement uncertainty and how the scale of biomes is appropriate for our national assessment.*

Zeri, M., Alvalá, R.C.S., Carneiro, R., Cunha-Zeri, G., Costa, J.M., Spatafora, L.R., Urbano, D., Vall-Llossera, M., Marengo, J., 2018. Tools for communicating agricultural drought over the Brazilian Semiarid using the soil moisture index. Water 10, 1421. https://doi.org/10.3390/w10101421

**Answers to Reviewer #3**

**Reviewer comment:** The authors have generally addressed my comments, although I wish they had indicated the specific changes made in their response. I still have a few comments/suggestions for the authors.Regarding my comment on soil moisture, the authors have not really addressed the matter, as they just proceeded with their analysis assuming that 5 cm soil moisture is a good proxy for droughts. I suggest the authors to at least include more references and explain the limitations of this approach. When can 5 cm soil moisture be a good proxy for soil water storage? When can it not?

**Response**: To investigate the coupling between surface and subsurface soil moisture is beyond the scope of this study. We use aggregated surface soil moisture values on a yearly basis across pixels with >25km of spatial resolution. With a national perspective the scale of our analysis is at the biome level. We observe differences in soil moisture trends across biomes that are consistent with trends of two sources of independent informacion, a) trends of vegetation productivity tand b) trends of emergency declaration by drought across biomes. These trends (in soil moisture decline, vegetation browning and emergency declarations by drought) are indicators of drought conditions across Brazil. We agree with the Reviewer in that satellite soil moisture is limited to represent only a component of the soil water storage at the soil surface. We have indicated the main limitations of satellite soil moisture in the introduction of the revised manuscript, we have also clarified that we do not want to represent the soil water storage only with surface soil moisture estimates.

**Reviewer comment:** Fig. 1 can be a little misleading, especially because the authors refer to it in line 52 after mentioning water scarcity. Fig. 1 has nothing to do with droughts. The Caatinga forest is a seasonal ecosystem and Fig.1 is how the Caatinga looks like every dry season.

**Response**: We have included two more elements in Fig. R1 striving to clarify that the Caatinga region is a vulnerable area to drought conditions not only because of it high contrast between dry (Figure R1a,) and wet season (Figure R1b), but also because there are administrative decisions about water management (e.g., intensifying land use across along channel networks, Figure R1b or, changing the course of rivers, Figure R1c) that consequently could contribute explaining the high number emergency declarations across this biome.

[Figure]

**Reviewer comment:** Figure R1 A perspective of the Caatinga forest during the dry season at the ground level (A), A perspective of land use in the Caatinga biome during the wet season at the landscape level (B). An example of human intervention to river course that has an impact on water availability across the region (C). Most importantly, it is not clear to me yet what soil moisture trends add to the analysis of droughts. What are the advantages compared to drought indexes? I believe that adding a specific subsection in the Results and Discussion may help both the reviewers and future readers.

**Response:** As stated in the introduction of our manuscript, soil moisture decline due to drought has a direct impact on agriculture, water security, and ecosystem services. Therefore, the lack of soil

moisture information could lead to inaccurate assessment of drought conditions, underestimation of drought impacts, and incomplete resilience and adaptation plans. In this paper we argue that soil moisture trends should be integrated into drought monitoring and early warning systems and soil conservation plans at national and local levels, which it is not so far in Brazil and in most countries in the world.